# Application of Selected Inoculant Producing Antifungal and Fibrinolytic Substances on Rye Silage with Different Wilting Time

Seong-Shin Lee [1,†] , Jeong-Seok Choi [1,†] , Dimas Hand Vidya Paradhipta [1,2] , Young-Ho Joo [1] , Hyuk-Jun Lee [1], Hyeon-Tak Noh [1], Dong-Hyeon Kim [3] and Sam-Churl Kim [1,*]

1   Division of Applied Life Science (BK21Four, Institute of Agriculture & Life Science), Gyeongsang National University, Jinju 52828, Korea; seongshin73@gmail.com (S.-S.L.); x47677104@gmail.com (J.-S.C.); dimazhand@gmail.com (D.H.V.P.); wn5886@gmail.com (Y.-H.J.); hyukjun0209@gmail.com (H.-J.L.); nht1647@gmail.com (H.-T.N.)
2   Faculty of Animal Science, Universitas Gadjah Mada, Yogyakarta 55281, Indonesia
3   Dairy Science Division, National Institute of Animal Science, Cheonan 31000, Korea; kimdh3465@gmail.com
*   Correspondence: kimsc@gnu.ac.kr; Tel.: +82-557721947; Fax: +82-557721949
†   These authors contributed equally to this work.

**Abstract:** This research was conducted to determine the effects of selected inoculant on the silage with different wilting times. The ryes were unwilted or wilted for 12 h. Each rye forage was ensiled for 100 d in quadruplicate with commercial inoculant (*Lactobacillus plantarum* sp.; LPT) or selected inoculant (*Lactobacillus brevis* 100D8 and *Leuconostoc holzapfelii* 5H4 at 1:1 ratio; MIX). In vitro dry matter digestibility and in vitro neutral detergent fiber digestibility were highest in the unwilted MIX silages ($p < 0.05$), and the concentration of ruminal acetate was increased in MIX silages ($p < 0.001$; 61.4% vs. 60.3%) by the increase of neutral detergent fiber digestibility. The concentration of ruminal ammonia-N was increased in wilted silages ($p < 0.001$; 34.8% vs. 21.1%). The yeast count was lower in the MIX silages than in the LPT silages ($p < 0.05$) due to a higher concentration of acetate in MIX silages ($p < 0.05$). Aerobic stability was highest in the wilted MIX silages ($p < 0.05$). In conclusion, the MIX inoculation increased aerobic stability and improved fiber digestibility. As a result of the wilting process, ammonia-N in silage decreased but ruminal ammonia-N increased. Notably, the wilted silage with applied mixed inoculant had the highest aerobic stability.

**Keywords:** bacteria inoculant; rumen fermentation; rye silage; wilting

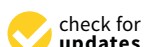



## 1. Introduction

The improvement of silage quality through bacterial additive has been commonly applied in the field. Even though improved ensiling processes, bacterial additives are not always guaranteed to reduce contamination by undesirable microbes after silo open [1]. Recently, many researchers revealed that lactic acid bacteria (LAB) can produce antifungal activity to reduce contamination by undesirable microbes on silage, especially after silo open [2–4]. In addition, acetate and propionate produced by LAB inhibit the growth of fungi during ensiling through cell wall breakage and mitochondria apoptosis, respectively [5,6]. According to previous studies, the application of LAB producing antifungal activity can reduce the growth of undesirable microbes and increase the aerobic stability of silage [3,4]. On the other hand, some forages are characterized by high structural carbohydrate and low soluble carbohydrate, which reduce either ensiling quality or digestibility [7]. In addition, an increased concentration of lignocellulose reduces the digestibility of silage in the rumen [8]. Several strains of LAB were reported to produce fibrinolytic enzymes such as esterase and xylanase, which could help to degrade structural carbohydrate and improve ruminal digestibility [8,9]. Application of LAB producing fibrinolytic activity on silage was reported to increase ruminal digestibility in many previous studies [4,8].

In South Korea, rye (*Secale cereale* L.) is a main winter forage for ruminant diet [10]. According to Kim et al. [11], rye has high adaptability and resistance in acid soil and also can be grown by double-cropping in rice paddies. In the field, rye produces high dry matter (DM) yield to maintain the requirement of a roughage source during winter season for ruminants. However, it also contains a high concentration of lignocellulose [12]. Until recently, studies concerned with scientific approaches to improving rye silage were limited compared to studies of other silages.

In our previous study, *Leuconostoc holzapfelii* 5H4 and *Lactobacillus brevis* 100D8 were isolated from rye silage and selected on the basis of their fibrinolytic and antifungal activity, respectively [2,13]. According to Kim et al. [2], *L. brevis* 100D8 has eight putative antibiotic biosynthesis genes that improve antifungal activity. In contrast, *Leuc. holzapfelii* 5H4 has genes encoding for esterase, cellulase, and xylanase protein, which can help to improve ruminal digestibility [13]. Paradhipta et al. [14] confirmed that *L. brevis* 100D8 produces antifungal activity, while *Leuc. holzapfelii* 5H4 produces fibrinolytic activity on rye silage. However, the effectiveness of selected LAB compared to commercial bacterial additive was not yet known. In addition, differences in moisture concentration of rye through the wilting process might also affect the improvement of silage quality by selected LAB. In the present study, *L. brevis* 100D8 and *Leuc. holzapfelii* 5H4 were mixed as silage inoculant to produce dual activities. In our hypothesis, the application of selected inoculant could produce better silage quality to improve ruminal digestibility and aerobic stability than commercial inoculant. Therefore, the present study was conducted to determine the effects of dual-activity inoculant on the fermentation quality and in vitro digestibility of rye silage at different wilting times compared to commercial inoculant.

## 2. Materials and Methods

### 2.1. Inoculant Preparation

The selected inoculant was produced at the animal research unit, Gyeongsang Animal Science and Technology (GAST), Jinju, South Korea. The selected inoculant consisted of *L. brevis* strain 100D8 and *Leuc. holzapfelii* strain 5H4 at a 1:1 ratio. Approximately 2 L of inoculant was mixed with 500 L of culture medium (glucose 14,000 g, soy peptone 1750 g, yeast extract 7000 g, magnesium sulfate 350 g, manganese sulfate 140 g, salt 350 g, sodium phosphate dibasic 2100 g, calcium carbonate 117 g, and NaOH for pH control). The inoculant was incubated at 35 °C for 12 h and then freeze-dried for 5 d.

### 2.2. Silage Production

Rye forage was grown in the animal research unit, Gyeongsang National University, Jinju, South Korea and harvested at the early dough stage (34.7% DM). The harvested rye was wilted under the sun as follows: (1) unwilted (UW) containing 34.7% DM; and (2) wilted for 12 h (WIL) containing 42.8% DM. Each rye forage was chopped into 3–5 cm lengths using a conventional harvester (BHC-90, BUHEUNG Machinery Ltd., Jinju, South Korea) and treated with different inoculants as follows: (1) commercial inoculant containing *Lactobacillus plantarum* sp. with application rate of $8.0 \times 10^5$ colony forming units (cfu)/g of fresh forage (LPT); and (2) selected inoculant consisting of *L. brevis* strain 100D8 and *Leuc. holzapfelii* strain 5H4 in 1:1 ratio with application rate of $7.5 \times 10^5$ cfu/g of fresh forage (MIX). All forages were ensiled into 20 L mini silos (5 kg) for 100 d in quadruplicate. Thus, a total of 16 silos were prepared in the present study. The rye forages before and after ensiling were sub-sampled at 500 g, respectively, to analyze chemical composition and in vitro digestibility. Also, 20 g of rye silage was sub-sampled and blended with 200 mL of sterile ultrapure water for 30 sec, and then filtered through 2 layers of cheesecloth to produce silage extract. The fresh silage extraction was used to analyze pH. Afterward, the silage extraction was stored at −70 °C until the analyses of ammonia-N, lactate, and volatile fatty acid (VFA).

### 2.3. Chemical Composition and In Vitro Digestiblity

The sub-sampled rye forage and silage (10 g) were dried at 105 °C for 24 h to measure the DM concentration. Approximately 200 g of each silage sub-sample was collected, dried at 60 °C for 48 h, and ground using a cutting mill (Shinmyung Electric Co., Ltd., Gimpo, South Korea) to pass through a 1 mm screen. The crude ash concentration (CA) was determined using a muffle furnace at 550 °C for 5 h. The crude protein (CP) and ether extract (EE) concentrations were analyzed by the Kjeldahl method (method 984.13) and the Soxhlet method (method 920.39), respectively. The neutral detergent fiber (NDF; method 2002.04) and acid detergent fiber (ADF; method 973.18) concentrations were determined using an Ankom[200] fiber analyzer (Ankom Technology, Macedon, NY, USA). All protocols for the CP, EE, NDF, and ADF analyses were described by AOAC [15]. Hemicellulose (HEMI) was determined by calculating the difference between the NDF and ADF. The in vitro digestibility of DM (IVDMD) and NDF (IVNDFD) was determined by following the method of Tilley and Terry [16] using Ankom[II] Daisy Incubators (Ankom Tech., Macedon, NY, USA).

### 2.4. Fermentation Characteristics

The pH and ammonia-N concentration were measured using pH meter (SevenEasy, Mettler Toledo, Switzerland) and colorimetry assay [17], respectively. The silage extract was centrifuged at $5645 \times g$ for 15 min, and the supernatant was used to measure the lactate and VFA concentrations using HPLC (L-2200; Hitachi, Tokyo, Japan) fitted with a UV detector (L-2400; Hitachi) and a column (Metacarb 87H; Varian, CA, USA) described by Adesogan et al. [18].

### 2.5. Microbial Enumerations

About 20 g of silage samples from each treatment were diluted with 180 mL of distilled water and macerated in a blender to obtain the silage extract for the enumeration of LAB, yeast, and mold. Considering the silage extract as the first dilution, serial dilutions were prepared and 100 μL aliquots of 3 consecutive dilutions ($10^{-4}$, $10^{-5}$, and $10^{-6}$) were plated in triplicate onto a selective agar medium. *Lactobacilli* MRS agar media (MRS; Difco, Detroit, MI, USA) was used to culture LAB, and potato dextrose agar (PDA; Difco, Detroit, MI, USA) was used for yeast and mold. The MRS agar plates were placed in a $CO_2$ incubator (Thermo Scientific, Waltham, MA, USA) at 30 °C for 72 h, while the PDA plates were incubated at 30 °C for 72 h in a normal incubator (Johnsam Corporation, Boocheon, South Korea). Visible colonies were counted from the plates, and the number of cfu was expressed per gram of silage. The microbial data were transformed to log10.

### 2.6. Rumen Fermentation

The animal care produce was approved by the Animal Ethical Committee of Gyeongsang National University, Jinju, South Korea. The rumen fluid was collected from two non-pregnant cannulated Hanwoo heifers before morning feeding. Their diets consisted of rice straw and commercial concentrate mix at a ratio of 8:2. The collected rumen fluid was composited and filtered through 2 layers of cheesecloth. A rumen buffer was prepared by mixing rumen fluid with anaerobic culture medium at a 1:2 ratio, as described by Adesogan et al. [19]. Dried samples of rye silage (0.5 g) were weighed into an incubation bottle with 40 mL of rumen buffer. Then, the incubation bottle was gassed with $CO_2$ and closed tightly to reach anaerobic conditions. Samples were incubated in triplicate for 0, 1, 3, 6, 12, 24, 48, and 72 h with 2 blanks for each time. The gas pressure was measured using a manometer pressure/vacuum gauge monitor (Fisher Scientific, Traceable, Friendswood, TX, USA) to calculate the rumen fermentation kinetics. The kinetics were generated from the gas pressure using the nonlinear regression procedure of SAS [20] to fit with the model of McDonald [21], as follows:

$$Y = A + B \left(1 - e^{-c(t-L)}\right) \text{ for } t > L$$

where A is the immediately degradable fraction, B is the potentially degradable fraction, A + B is the total degradable fraction, C is the fractional degradation rate, L is lag phase, and t is time of incubation (h).

The bottles incubated for 72 h were opened and transferred to a 50 mL conical tube to separate the remaining sample and supernatant (rumen buffer) through centrifugation at $2568\times g$ for 15 min (Supra 21k, Hanil Electric Corporation, Seoul, South Korea, with rotor A50S-6C No.6). The supernatant was used to analyze rumen fermentation characteristics such as the pH, ammonia-N, and VFA. The procedure for the analyses of pH, ammonia-N, and VFA were the same as described in the previous section.

### 2.7. Statistical Analysis

This experiment had a completely randomized design with a 2 (wilting; UW vs. WIL) × 2 (inoculant; LPT vs. MIX) factorial arrangement of the treatments. All data on the chemical composition, fermentation characteristics, microbe counts, and temperature of the silages were analyzed using PROC MIXED of SAS [20] and a model containing the day, inoculant, and interactions of these terms. The model was $Y_{ijk} = \mu + \alpha_i + \beta_j + (\alpha\beta)_{ij} + e_{ijk}$, where $Y_{ijk}$ = response variable, $\mu$ = overall mean, $\alpha_i$ = effect of time treatment, $\beta_j$ = effect of inoculant treatment, $(\alpha\beta)_{ij}$ = the interaction effect of time and inoculant, and $e_{ijk}$ = error term. Mean separation was performed using a Tukey's test. Significant differences were declared at $p < 0.05$.

### 3. Results

### 3.1. Chemical Compositions and In Vitro Digestibility

Wilted rye forage had a higher DM concentration ($p < 0.001$; 42.8% vs. 34.7%) than unwilted rye forage (Table 1). The mean concentrations of CP, EE, CA, NDF, ADF, and HEMI in rye forages were 5.98%, 1.98%, 5.16%, 72.0%, 42.3%, and 29.7%, respectively. The UW silages had a higher concentration of DM ($p < 0.001$; 32.9% vs. 40.0%) than WIL silages (Table 2). The MIX silages had higher concentrations of NDF ($p < 0.001$; 73.7% vs. 70.9%), HEMI ($p = 0.005$; 31.8% vs. 29.4%), IVDMD ($p = 0.005$; 55.2% vs. 51.2%), and IVNDFD ($p = 0.045$; 45.9% vs. 43.7%) than LPT silages.

**Table 1.** Effects of wilting and inoculants on chemical compositions of rye forage before ensiling (%, DM).

| Item [1] | | UW | | | WIL | | SEM |
|---|---|---|---|---|---|---|---|
| | | LPT | MIX | | LPT | MIX | |
| DM | | 34.7 [b] | 34.6 [b] | | 42.8 [a] | 42.8 [a] | 0.891 |
| CP | | 5.88 | 5.84 | | 6.13 | 6.06 | 0.247 |
| EE | | 1.83 | 2.17 | | 1.97 | 1.96 | 0.258 |
| CA | | 5.05 | 5.27 | | 5.13 | 5.18 | 0.181 |
| NDF | | 72.3 | 71.9 | | 72.5 | 71.3 | 1.006 |
| ADF | | 42.7 | 41.8 | | 42.9 | 41.7 | 1.053 |
| HEMI | | 29.6 | 30.0 | | 29.6 | 29.6 | 0.436 |
| Contrast | DM | CP | EE | CA | NDF | ADF | HEMI |
| WT | <0.001 | 0.134 | 0.811 | 0.924 | 0.878 | 0.977 | 0.214 |
| INO | 0.863 | 0.694 | 0.263 | 0.162 | 0.165 | 0.103 | 0.585 |
| WT × INO | 0.888 | 0.928 | 0.241 | 0.366 | 0.469 | 0.774 | 0.639 |

[1] UW, un-wilting; WIL, wilting for 12 h; LPT, *L. plantarum* ($8.0 \times 10^5$ cfu/g); MIX, mixture of *Lactobacillus brevis* strain 100D8 and *Leuconostoc holzapfelii* strain 5H4 at 1:1 ratio ($7.5 \times 10^5$ cfu/g); DM, dry matter; CP, crude protein; EE, ether extract; CA, crude ash; NDF, neutral detergent fiber; ADF, acid detergent fiber; HEMI, hemicellulose; WT, wilting effect; INO, inoculant effect; WT×INO, interaction effect between wilting and inoculant; SEM, standard error of the mean. [a,b] Means in the same row with different superscripts differ significantly ($p < 0.05$).

**Table 2.** Effects of wilting and inoculants on chemical compositions and in vitro digestibility of rye silage ensiled for 100 d (%, DM).

| Item [1] | UW | | | WIL | | SEM |
|---|---|---|---|---|---|---|
| | LPT | MIX | | LPT | MIX | |
| DM | 33.3 [b] | 32.5 [b] | | 40.5 [a] | 39.5 [a] | 0.922 |
| CP | 7.07 | 6.94 | | 6.96 | 7.13 | 0.428 |
| EE | 3.86 | 3.61 | | 3.84 | 3.35 | 0.376 |
| CA | 5.39 | 5.50 | | 5.33 | 5.40 | 0.102 |
| NDF | 71.5 [b] | 73.8 [a] | | 70.2 [b] | 73.5 [a] | 0.802 |
| ADF | 41.9 | 41.6 | | 41.4 | 42.2 | 0.903 |
| HEMI | 30.0 | 32.2 | | 28.8 | 31.3 | 1.117 |
| IVDMD | 52.5 [ab] | 56.7 [a] | | 49.8 [b] | 53.6 [ab] | 1.657 |
| IVNDFD | 45.1 [ab] | 46.9 [a] | | 42.2 [b] | 44.9 [ab] | 1.215 |

| Contrast | DM | CP | EE | CA | NDF | ADF | HEMI | IVDMD | IVNDFD |
|---|---|---|---|---|---|---|---|---|---|
| WT | <0.001 | 0.837 | 0.537 | 0.176 | 0.081 | 0.915 | 0.364 | 0.013 | 0.028 |
| INO | 0.102 | 0.912 | 0.119 | 0.112 | <0.001 | 0.696 | 0.005 | 0.005 | 0.045 |
| WT × INO | 0.849 | 0.494 | 0.590 | 0.724 | 0.251 | 0.301 | 0.822 | 0.874 | 0.564 |

[1] UW, un-wilting; WIL, wilting for 12 h; LPT, *L. plantarum* ($8.0 \times 10^5$ cfu/g); MIX, mixture of *Lactobacillus brevis* strain 100D8 and *Leuconostoc holzapfelii* strain 5H4 at 1:1 ratio ($7.5 \times 10^5$ cfu/g); DM, dry matter; CP, crude protein; EE, ether extract; CA, crude ash; NDF, neutral detergent fiber; ADF, acid detergent fiber; HEMI, hemicellulose; WT, wilting effect; INO, inoculant effect; WT×INO, interaction effect between wilting and inoculant; SEM, standard error of the mean. [a,b] Means in the same row with different superscripts differ significantly ($p < 0.05$).

### 3.2. Fermentation Characteristics

The MIX silages had a higher pH value ($p < 0.001$; 4.10 vs. 4.62) and concentration of acetate ($p < 0.001$; 4.48% vs. 0.88%) than LPT silage (Table 3). However, the MIX silages had lower lactate to acetate ratio ($p < 0.001$; 0.10 vs. 7.41). The interaction effects between wilting and inoculant were shown in ammonia-N ($p = 0.010$); MIX was only effective in reducing ammonia-N on WIL silage. The interaction effects between wilting and inoculant were also presented in lactate ($p = 0.012$) and propionate ($p < 0.001$). It could be seen that only on UW silage, LPT and MIX were effective in increasing ($p < 0.05$) concentrations of lactate and propionate, respectively. The concentration of butyrate was not detected in any of the treatments.

**Table 3.** Effects of wilting and inoculants on fermentation characteristics of rye silage ensiled for 100 d.

| Item [1] | UW | | | WIL | | SEM |
|---|---|---|---|---|---|---|
| | LPT | MIX | | LPT | MIX | |
| pH | 4.06 [b] | 4.58 [a] | | 4.14 [b] | 4.65 [a] | 0.043 |
| Ammonia-N, % DM | 0.026 [bc] | 0.041 [a] | | 0.022 [c] | 0.029 [b] | 0.003 |
| Lactate, % DM | 7.40 [a] | 0.24 [c] | | 5.30 [b] | 0.61 [c] | 0.647 |
| Acetate, % DM | 1.11 [b] | 4.74 [a] | | 0.65 [b] | 4.22 [a] | 1.062 |
| Propionate, % DM | ND [b] | 0.05 [a] | | ND [b] | 0.01 [b] | 0.004 |
| Butyrate, % DM | ND | ND | | ND | ND | N/A |
| Lactate:acetate ratio | 6.66 [a] | 0.05 [b] | | 8.15 [a] | 0.14 [b] | 0.705 |

| Contrast | pH | Ammonia-N | Lactate | Acetate | Propionate | Butyrate | L:A |
|---|---|---|---|---|---|---|---|
| WT | 0.016 | <0.001 | 0.055 | 0.198 | <0.001 | N/A | 0.087 |
| INO | <0.001 | <0.001 | <0.001 | <0.001 | <0.001 | N/A | <0.001 |
| WT × INO | 0.709 | 0.010 | 0.012 | 0.963 | <0.001 | N/A | 0.116 |

[1] UW, un-wilting; WIL, wilting for 12 h; LPT, *L. plantarum* ($8.0 \times 10^5$ cfu/g); MIX, mixture of *Lactobacillus brevis* strain 100D8 and *Leuconostoc holzapfelii* strain 5H4 at 1:1 ratio ($7.5 \times 10^5$ cfu/g); L:A, lactate to acetate ratio; WT, wilting effect; INO, inoculant effect; WT×INO, interaction effect between wilting and inoculant; SEM, standard error of the mean; ND, <0.01% DM; N/A, not applicable. [a–c] Means in the same row with different superscripts differ significantly ($p < 0.05$).

### 3.3. Microbial Counts

The MIX silages had lower yeast count ($p$ <0.001; 4.79 log10 cfu/g vs. 5.75 log10 cfu/g) than LPT silage (Table 4). The interaction effects between wilting and inoculant were shown in LAB ($p$ = 0.015) and aerobic stability ($p$ < 0.001); MIX was effective in increasing ($p$ < 0.05) aerobic stability only on WIL silage. Mold was not detected in any of the treatments.

**Table 4.** Effects of wilting and inoculants on microbial counts and aerobic stability of rye silage ensiled for 100 d.

| Item [1] | UW | | WIL | | SEM |
|---|---|---|---|---|---|
| | LPT | MIX | LPT | MIX | |
| LAB, $\log_{10}$ cfu/g | 5.77 [c] | 7.84 [a] | 5.57 [c] | 6.72 [b] | 0.322 |
| Yeast, $\log_{10}$ cfu/g | 5.55 [a] | 4.84 [b] | 5.94 [a] | 4.73 [b] | 0.270 |
| Mold, $\log_{10}$ cfu/g | ND | ND | ND | ND | N/A |
| Aerobic stability, h | 39.4 [c] | 430.7 [b] | 53.7 [c] | 722.0 [a] | 12.77 |

| Contrast | LAB | Yeast | Mold | Aerobic stability |
|---|---|---|---|---|
| WT | 0.002 | 0.713 | N/A | <0.001 |
| INO | <0.001 | <0.001 | N/A | <0.001 |
| WT × INO | 0.015 | 0.156 | N/A | <0.001 |

[1] UW, un-wilting; WIL, wilting for 12 h; LPT, *L. plantarum* ($8.0 \times 10^5$ cfu/g); MIX, mixture of *Lactobacillus brevis* strain 100D8 and *Leuconostoc holzapfelii* strain 5H4 at 1:1 ratio ($7.5 \times 10^5$ cfu/g); WT, wilting effect; INO, inoculant effect; WT×INO, interaction effect between wilting and inoculant; SEM, standard error of the mean; ND, <4.0 log10 cfu/g. [a–c] Means in the same row with different superscripts differ significantly ($p$ < 0.05).

### 3.4. Rumen Fermentation Kinetics

The MIX silages had a higher immediately fermentable fraction ($p$ < 0.001; 0.38 mL/g vs. 0.25 mL/g) but a lower fractional fermentation rate ($p$ = 0.004; 0.04 vs. 0.05) than LPT silages (Table 5). The interaction effects between wilting and inoculant were shown in the potentially fermentable fraction ($p$ = 0.017) and the total fermentable fraction ($p$ = 0.020); LPT was effective in increasing ($p$ < 0.05) the potentially fermentable fraction and the total fermentable fraction on WIL silage. The interaction effects between wilting and inoculant were also presented in the lag phase ($p$ = 0.021), and it could be seen that LPT was effective in decreasing ($p$ < 0.05) lag phase only on WIL silage.

**Table 5.** Effects of wilting and inoculants on rumen fermentation kinetics of rye silage incubated with rumen buffer for 72 h.

| Item [1] | UW | | WIL | | SEM |
|---|---|---|---|---|---|
| | LPT | MIX | LPT | MIX | |
| A, mL/g of DM | 0.26 [b] | 0.37 [a] | 0.23 [b] | 0.38 [a] | 0.027 |
| B, mL/g of DM | 12.5 [b] | 14.3 [ab] | 16.3 [a] | 15.3 [a] | 0.825 |
| A+B, mL/g of DM | 12.8 [b] | 14.7 [ab] | 16.5 [a] | 15.7 [a] | 0.838 |
| C, %/h | 0.06 [a] | 0.04 [ab] | 0.03 [b] | 0.04 [b] | 0.006 |
| L, h | 2.67 [a] | 2.41 [ab] | 1.74 [b] | 2.60 [ab] | 0.342 |

| Contrast | A | B | A + B | C | L |
|---|---|---|---|---|---|
| WT | 0.575 | 0.001 | 0.001 | 0.004 | 0.097 |
| INO | <0.001 | 0.429 | 0.308 | 0.351 | 0.166 |
| WT × INO | 0.162 | 0.017 | 0.020 | 0.052 | 0.021 |

[1] UW, un-wilting; WIL, wilting for 12 h; LPT, *L. plantarum* ($8.0 \times 10^5$ cfu/g); MIX, mixture of *Lactobacillus brevis* strain 100D8 and *Leuconostoc holzapfelii* strain 5H4 at 1:1 ratio ($7.5 \times 10^5$ cfu/g); A, the immediately fermentable fraction; B, the potentially fermentable fraction; A + B, the total fermentable fraction; C, the fractional fermentation rate; L, the lag phase; WT, wilting effect; INO, inoculant effect; WT×INO, interaction effect between wilting and inoculant; SEM, standard error of the mean. [a,b] Means in the same row with different superscripts differ significantly ($p$ < 0.05).

### 3.5. Rumen Fermentation Characteristics

The WIL silages had higher concentrations of ammonia-N ($p < 0.001$; 34.8 vs. 21.1 mg/dL), iso-butyrate ($p = 0.022$; 1.34% vs. 1.21% molar), butyrate ($p = 0.015$; 13.9% vs. 13.2% molar), and valerate ($p = 0.018$; 1.56% vs. 1.16% molar) than UW silage (Table 6). The MIX silages had a higher concentration of acetate ($p < 0.001$; 61.4% vs. 60.3% molar) and iso-valerate ($p = 0.020$; 3.09% vs. 2.69% molar) than LPT silage. The interaction effects between wilting and inoculant were shown in total VFA ($p = 0.026$), the concentration of which was effectively improved ($p < 0.05$) by MIX only on UW silage. The interaction effects between wilting and inoculant were shown in propionate ($p = 0.007$), and acetate to propionate ratio ($p = 0.005$). It could be seen that only on UW silage, LPT could increase ($p < 0.05$) propionate and decrease ($p < 0.05$) the ratio of acetate to propionate effectively.

**Table 6.** Effects of wilting and inoculants on rumen pH, ammonia-N, and volatile fatty acids of rye silage incubated with rumen buffer for 72 h.

| Item [1] | UW | | | WIL | | SEM |
|---|---|---|---|---|---|---|
| | LPT | MIX | | LPT | MIX | |
| pH | 6.03 | 5.97 | | 5.94 | 5.98 | 0.043 |
| Ammonia-N, mg/dL | 21.4 [b] | 20.7 [b] | | 34.5 [a] | 35.0 [a] | 0.599 |
| Total VFA, m$M$/L | 71.2 [b] | 104.4 [a] | | 89.0 [ab] | 90.6 [ab] | 6.505 |
| Acetate, % of molar | 60.1 [b] | 61.5 [a] | | 60.4 [b] | 61.3 [a] | 0.214 |
| Propionate, % of molar | 22.1 | 19.5 | | 19.3 | 19.1 | 0.588 |
| Iso-butyrate, % of molar | 1.16 | 1.26 | | 1.32 | 1.35 | 0.078 |
| Butyrate, % of molar | 13.1 | 13.2 | | 14.2 | 13.6 | 0.427 |
| Iso-valerate, % of molar | 2.46 | 2.96 | | 2.92 | 3.21 | 0.235 |
| Valerate, % of molar | 0.99 | 1.33 | | 1.65 | 1.46 | 0.228 |
| Acetate:propionate ratio | 2.74 [c] | 3.15 [ab] | | 3.10 [b] | 3.24 [a] | 0.034 |

| Contrast | pH | NH$_3$-N | TVFA | AC | PR | IBU | BU | IVA | VA | A:P |
|---|---|---|---|---|---|---|---|---|---|---|
| WT | 0.121 | <0.001 | 0.686 | 0.804 | 0.001 | 0.022 | 0.015 | 0.028 | 0.018 | <0.001 |
| INO | 0.608 | 0.881 | 0.019 | <0.001 | 0.005 | 0.188 | 0.311 | 0.020 | 0.568 | <0.001 |
| WT × INO | 0.080 | 0.210 | 0.026 | 0.056 | 0.007 | 0.503 | 0.175 | 0.455 | 0.076 | 0.005 |

[1] UW, un-wilting; WIL, wilting for 12 h; LPT, *L. plantarum* ($8.0 \times 10^5$ cfu/g); MIX, mixture of *Lactobacillus brevis* strain 100D8 and *Leuconostoc holzapfelii* strain 5H4 at 1:1 ratio ($7.5 \times 10^5$ cfu/g); NH$_3$-N, ammonia-N; TVFA, total volatile fatty acid; AC, acetate; PR, propionate; IBU, iso-butyrate; BU, butyrate; IVA, iso-valerate; VA, valerate; A:P, acetate to propionate ratio; WT, wilting effect; INO, inoculant effect; WT×INO, interaction effect between wilting and inoculant; SEM, standard error of the mean. [a–c] Means in the same row with different superscripts differ significantly ($p < 0.05$).

### 4. Discussion

According to previous studies, generally, concentrations of CP, EE, CA, NDF, ADF, and HEMI from rye forages were 6.48–10.6%, 1.49–2.22%, 5.13–6.11%, 70.5–74.4%, 45.9–46.2%, and 24.8–48.2%, respectively [14,22]. The results of the present study were slightly different from previous studies. This difference could have occurred because of different seed quantities and harvest times [14]. In general, hetero types of inoculant applications can result in a gradual decrease of silage pH, which also leads to enhance proteolysis and water-soluble carbohydrate (WSC) loss. Zeng et al. [23] reported that the increase of NDF could occur due to the decrease of WSC or CP concentration by undesired bacteria. This explanation was consistent with the results of the present study showing that NDF concentration increased in MIX silage. In the present study, in vitro digestibility decreased with wilting time. Gomes et al. [24] and Chen et al. [25] reported that the in vitro digestibility of wilted silage could be decreased by the loss of soluble nutrients and lead to the increase of cell external wall density. On the other hand, cellulase and xylanase are well-known to cleave the binding of cellulose, HEMI, and glucans, thereby increasing the fiber digestibility [26]. In particular, the lignocellulose is difficult to be degraded by bacteria [27,28]. However, fibrinolytic enzymes such as ferulate esterases could enhance the degradation of lignocellulose into pentoses and lead to an increase in acetic acid concentration of silage [29].

Degrading lignocellulose could increase fiber digestibility in the rumen [4,8,13]. Again, Kim et al. [13] reported that the *Leuc. holzapfelii* strain 5H4 used in the MIX inoculant produced several fibrinolytic enzymes such as esterase, cellulose, and xylanase. Therefore, the increased IVDMD and IVNDFD in MIX silages are most likely the result of the *Leuc. holzapfelii* strain 5H4 application.

The *L. plantarum* used in the present study is known as a homofermentative LAB that produces lactate as a main metabolite product [30]. Lactate can decrease silage pH rapidly in the initial stage of the ensiling period, which can help reduce nutrient loss by the inhibition of undesirable microbes [18]. The present study also showed similar results of lower pH with higher lactate concentration in silage treated with the LPT application. Ammonia-N in silage, as a by-product of proteolysis, inhibits the decrease of silage pH [30]. The lower ammonia-N attributable to LPT application in the present study also could support lower pH in LPT silage (Table 3).

*Leuc. holzapfelii* and *L. brevis* were confirmed as heterofermentative LABs [2,13]. They could convert lactate into acetate and propionate, which are known as antifungal substances [31]. In addition, *L. brevis* 100D8 has genes encoding lanthionine synthetase C-like protein, which has a role as an antifungal substance to inhibit undesirable microbes [2]. In addition, Paul and Donk [32] also reported that lanthionine can inhibit the growth of silage mold species such as *Aspergillus, Penicillium, and Fusarium*. In the present study, the lower yeast count in MIX silages might be a result of *Leuc. holzapfelii* and *L. brevis* applications. Ranjit and Kung [33] reported that yeast assimilates lactic acid of silage during aerobic exposure, and leads to the deterioration of aerobic spoilage by the growth of mold. Similarly, in the present study, the lower aerobic stability in LPT silages could be attributable to higher yeast counts and acetic acid concentrations in those silages. Generally, wilting could lead to decreased aerobic stability due to the growth of undesirable microbes [34]. However, some studies have reported that a wilting process of less than 10 h or until 30–40% of DM had beneficial effects on the fermentation quality and aerobic stability of silages [35–37]. In addition, Hu et al. [38] also reported that moderately high DM silage, compared to normal DM silage (40.6% vs. 33.1%), had an improved LAB count and aerobic stability. The wilting effects on aerobic stability observed in the present study were in agreement with these previous studies.

Adesogan [39] reported that fibrinolytic enzymes such as cellulase, xylanase, and esterase can convert plant cell walls into mono- or oligosaccharide. In the present study, the reason for the increased immediately fermentable fraction in MIX silage might have been the *Leuc. holzapfelii* strain 5H4 application, which enhanced the secretion of fibrinolytic enzymes. Morgan et al. [40] reported that the rumen nitrogen of steers could be increased if they were fed high-quality ryegrass silage. Ruminal ammonia-N is the main nitrogen source for microbial protein synthesis in the rumen [41]. Moreover, the synthesized microbial protein in the rumen provides amino acids more efficiently than the feed protein fed to the ruminants [42]. Charmley and Veira [43] reported that the protease activity in alfalfa silage is inhibited by the wilting process. In the present study, the ammonia-N concentration of silage was also decreased by wilting, which might be caused by the inhibition of proteolysis (Table 3). The increase in ruminal ammonia-N concentration obtained with wilted silage in the present study could be supported by this evidence. The total VFA is known as the main energy source for the growth of ruminants and provides about 80% of the energy for ruminants [44]. Wan et al. [45] reported that total VFA concentration was positively correlated with the total fermentable fraction. This explanation is supported by the results of the present study for unwilted LPT silage, which had a low total fermentable fraction with low total VFA concentration. Meller et al. [46] reported that the concentration of acetate in the rumen could be increased by increasing cellulolytic bacteria activity and NDF digestibility. The present study obtained similar results with the finding that the application of *Leuc. holzapfelii* strain 5H4, fibrinolytic enzyme producing bacteria, increased ruminal acetate concentration through increased IVNDFD (Table 2).

## 5. Conclusions

In conclusion, the antifungal and fibrinolytic effects on rye silage from mixed inoculant consisting of *Leuconostoc holzapfelii* 5H4 and *Lactobacillus brevis* 100D8 were confirmed by the results of increased IVDMD, IVNDFD, acetate, LAB, aerobic stability, and ruminal total VFA as well as decreased lactate and yeast. With the wilting process, ruminal ammonia-N was increased by low ammonia-N concentrations in silage. Additionally, aerobic stability was also increased by the wilting process, and was its highest with wilted silage treated with the mixed inoculant application.

**Author Contributions:** Conceptualization, S.-C.K. and S.-S.L.; methodology, S.-S.L., J.-S.C., D.H.V.P., H.-J.L., Y.-H.J., H.-T.N., and S.-C.K.; software, S.-S.L. and D.-H.K.; formal analysis, S.-S.L. and J.-S.C.; validation, S.-S.L., J.-S.C., D.H.V.P., H.-J.L., Y.-H.J., and H.-T.N.; supervision, S.-C.K. and D.-H.K.; data curation, S.-S.L., J.-S.C., D.H.V.P., H.-J.L., and Y.-H.J.; resources, S.-S.L. and S.-C.K.; writing—original draft preparation, S.-S.L. and J.-S.C.; writing—review and editing, S.-S.L., J.-S.C., D.H.V.P., D.-H.K., and S.-C.K. All authors have read and agreed to the published version of the manuscript.

**Funding:** This research was supported (Project No. 321083-05-1-HD040) by IPET (Korea Institute of Planning and Evaluation for Technology in Food, Agriculture, Forestry and Fisheries), and Ministry of Agriculture, Food and Rural Affairs, Republic of Korea.

**Institutional Review Board Statement:** The animal care and procedure to maintain cannulated heifers was approved by animal ethical committee of Gyeongsang National University, Jinju, South Korea (GNU-191011-E0050).

**Informed Consent Statement:** Not applicable.

**Data Availability Statement:** Data are available on request from the corresponding author with justifiable reason.

**Conflicts of Interest:** The authors declare no conflict of interest.

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
