# Peer review of "Application of Selected Inoculant Producing Antifungal and Fibrinolytic Substances on Rye Silage with Different Wilting Time"

_processes, doi:10.3390/pr9050879_

Round 1

Reviewer 1 Report

Presented manuscript is on high scientific level. The manuscript authors the introduction section presented the current state of knowledge on the experimental design. The topic is a new, as well present a very important chemical attributes and method of preparing of rye silage quality .

The Introduction section includes all necessary information about examined objects and problems.

Materials and method section is not a raised my doubts. All details are good described.

The discussion section presents a good comparison of the obtained results with other results available in the data basis.

Presented conclusions are corresponding with obtained results.

General opinion:  After carefully manuscript reading, I think, that presented experiment is a valuable. In my opinion Manuscript should be accept. It is perfect for Processes  journal publication.

Author Response

Thanks for your constructive comments. Your comments help us to improve this manuscript. The responses to comments are mentioned below:

Presented manuscript is on high scientific level. The manuscript authors the introduction section presented the current state of knowledge on the experimental design. The topic is a new, as well present a very important chemical attributes and method of preparing of rye silage quality .

The Introduction section includes all necessary information about examined objects and problems.

Materials and method section is not a raised my doubts. All details are good described.

The discussion section presents a good comparison of the obtained results with other results available in the data basis.

Presented conclusions are corresponding with obtained results.

General opinion:  After carefully manuscript reading, I think, that presented experiment is a valuable. In my opinion Manuscript should be accept. It is perfect for Processes  journal publication.

Response: Thanks for your interest and comments on this manuscript.

Reviewer 2 Report

The authors report the effects of an inoculant on rye silage with the aim of decreasing contamination after silo open. The results are clearly reported and introduction, methods and discussion are well supported with references, however there are some minor comments I’d suggest the authors should revise before publishing.

  • Section 2.1: the authors report that freeze-drying took place for five days. I understand such a long time is needed probably due to the amount of volume to lyophilise. However, it looks expensive in terms of energy and cost. Have the authors tried another drying method? Probably decreasing volumes or dividing the volume in different aliquots results more beneficial and also speeds up the process.
  • Section 2.2.: the authors report that harvested rye was wilted under the sun. Are there any alternatives in case the sun does not come out? Does it need to rely on the weather? Would it be ok even if it’s cloudy/rainy?
  • The authors report through the text “Leuc. holzapfelii”, but it should be “L. holzapfelii”.
  • The results are clearly presented in tables, however it results a bit tedious and difficult to follow as the reader ends overwhelmed with so many letters and numbers. I’d suggest reporting any of the results in graphs instead of tables. As it’d make it an easier way to understand and follow the authors’ research.

Author Response

Thanks for your constructive comments. Your comments help us to improve this manuscript. The responses to comments are mentioned below:

Section 2.1: the authors report that freeze-drying took place for five days. I understand such a long time is needed probably due to the amount of volume to lyophilise. However, it looks expensive in terms of energy and cost. Have the authors tried another drying method? Probably decreasing volumes or dividing the volume in different aliquots results more beneficial and also speeds up the process.

 Response: Thanks for your constructive comments. The main reason we used freeze-drying technique is to minimize the decrease of count number from our inoculant due to storage process. Freezing-drying is the best option to reduce the volume without significantly impact on microbial count of inoculant. Yes, it might be expensive. However, inoculant in the present study was produced in industrial scale, which could reduce the cost significantly if compared to laboratory scale.

Section 2.2.: the authors report that harvested rye was wilted under the sun. Are there any alternatives in case the sun does not come out? Does it need to rely on the weather? Would it be ok even if it’s cloudy/rainy?

  Response: Thanks for your comments. This study was conducted with consideration of practical condition in the field/farm. By this reason, we wilted the rye under the sun, which was similar with what farmer did in the field. Therefore, the results of this study were applicative for farmers. As you mentioned, the climatic condition is one of the key factors for silage quality. When it’s cloudy, the wilting time might be increased. And, when it rains, it’s not recommended to harvest the forages for silages.

The authors report through the text “Leuc. holzapfelii”, but it should be “L. holzapfelii”.

 Response: In general, “L.” represent Lactobacillus. It could be misunderstood if we only used "L." since there are two species of LAB in this study (Lactobacillus and Leuconostoc). In addition, "Leuc." was used commonly in various studies (Wittouck et al., 2019 at mSystems; Paradhipta et al., 2020 at Asian-Australasian journal of animal sciences; Woo et al., 2021 at Journal of Applied Microbiology).

The results are clearly presented in tables, however it results a bit tedious and difficult to follow as the reader ends overwhelmed with so many letters and numbers. I’d suggest reporting any of the results in graphs instead of tables. As it’d make it an easier way to understand and follow the authors’ research.

 Response: Thanks for your opinion. This study was conducted by factorial design. With factorial design, we would like to keep the table instead of graphs because the reader could be get more information regarding treatment effect such as wilting effect, inoculant effect, or interaction effect among wilting and inoculant.